# Generating Adversarial Examples with Adversarial Networks

## Abstract

Deep neural networks (DNNs) have been found to be vulnerable to adversarial examples resulting from adding small-magnitude perturbations to inputs. Such adversarial examples can mislead DNNs to produce adversary-selected results. Different attack strategies have been proposed to generate adversarial examples, but how to produce them with high perceptual quality and more efficiently requires more research efforts. In this paper, we propose AdvGAN to generate adversarial examples with generative adversarial networks (GANs), which can learn and approximate the distribution of original instances. For AdvGAN, once the generator is trained, it can generate adversarial perturbations efficiently for any instance, so as to potentially accelerate adversarial training as defenses. We apply AdvGAN in both semi-whitebox and black-box attack settings. In semi-whitebox attacks, there is no need to access the original target model after the generator is trained, in contrast to traditional white-box attacks. In black-box attacks, we dynamically train a distilled model for the black-box model and optimize the generator accordingly. Adversarial examples generated by AdvGAN on different target models have high attack success rate under state-of-the-art defenses compared to other attacks. Our attack has placed the first with 92.76% accuracy on a public MNIST black-box attack challenge (Mądry et al., 2017b).

## 1    Introduction

Deep Neural Networks (DNNs) have achieved great successes in a variety of applications ranging from image recognition (Krizhevsky et al., 2012; He et al., 2016) to speech processing (Hinton et al., 2012) and from robotics training (Levine et al., 2016) to medical diagnostics (Ciresan et al., 2012). However, recent work has demonstrated that DNNs are vulnerable to adversarial perturbations (Szegedy et al., 2014; Goodfellow et al., 2015). An adversary can add small-magnitude perturbations to inputs and generate adversarial examples to mislead DNNs. Such maliciously perturbed instances can cause the learning system to misclassify them into either a maliciously-chosen target class (in a targeted attack) or classes that are different from the ground truth (in an untargeted attack). Different algorithms have been proposed for generating such adversarial examples, such as the fast gradient sign method (FGSM) (Goodfellow et al., 2015) and optimization-based methods (Opt.) (Carlini & Wagner, 2017a; Liu et al., 2017).

Most of the the current attack algorithms (Carlini & Wagner, 2017a; Liu et al., 2017) rely on optimization schemes with simple pixel space metrics, such as $L_\infty$ distance from a benign image, to encourage visual realism. To generate more perceptually realistic adversarial examples, in this paper, we propose to train a feed-forward network to generate perturbations such that the resulting example must be realistic according to a discriminator network. We apply generative adversarial networks (GANs) (Goodfellow et al., 2014) to produce adversarial examples in both the semi-whitebox and black-box settings. As conditional GANs are capable of producing high-quality images (Isola et al., 2017), we apply a similar paradigm to produce perceptually realistic adversarial instances. We name our method AdvGAN.

Note that in the previous white-box attacks, such as FGSM and optimization methods, the adversary needs to have white-box access to the architecture and parameters of the model all the time. However, by deploying AdvGAN, once the feed-forward network is trained, it can instantly pro-

duce adversarial perturbations for any input instances without requiring access to the model itself anymore. We name this attack setting *semi-whitebox*.

To evaluate the effectiveness of our attack strategy AdvGAN, we first generate adversarial instances based on AdvGAN and other attack strategies on different target models. We then apply the state-of-the-art defenses to defend against these generated adversarial examples (Goodfellow et al., 2015; Tramèr et al., 2017a; Mądry et al., 2017a). We evaluate these attack strategies in both semi-whitebox and black-box settings. We show that adversarial examples generated by AdvGAN can achieve a high attack success rate, potentially due to the fact that these adversarial instances appear closer to real instances compared to other recent attack strategies.

Our contributions are listed as follows.

- Different from the previous optimization-based methods, we train a conditional adversarial network to directly produce adversarial examples, which are both perceptually realistic and achieve state-of-the-art attack success rate against different target models.
- We show that AdvGAN can attack black-box models by training a distilled model. We propose to dynamically train the distilled model with query information and achieve high black-box attack success rate and targeted black-box attack, which is difficult to achieve for transferability-based black-box attacks.
- We use the state-of-the-art defense methods to defend against adversarial examples and show that AdvGAN achieves much higher attack success rate under current defenses.
- We apply AdvGAN on Mądry et al.'s MNIST challenge (2017a) and achieve 88.93% accuracy on the published robust model in the semi-whitebox setting and 92.76% in the black-box setting, which wins the top position in the challenge (Mądry et al., 2017b).

## 2 RELATED WORK

Here we review recent work on adversarial examples and generative adversarial networks.

**Adversarial Examples** A number of attack strategies to generate adversarial examples have been proposed in the white-box setting, where the adversary has full access to the classifier (Szegedy et al., 2014; Goodfellow et al., 2015; Carlini & Wagner, 2017a; Moosavi-Dezfooli et al., 2015; Papernot et al., 2016; Biggio et al., 2013; Kurakin et al., 2016). Goodfellow et al. propose the fast gradient sign method (FGSM), which applies a first-order approximation of the loss function to construct adversarial samples. Formally, given an instance $x$, an adversary generates adversarial example $x_A = x + \eta$ with $L_\infty$ constraints in the untargeted attack setting as $\eta = \epsilon \cdot \text{sign}(\nabla_x \ell_f(x, y))$, where $\ell_f(\cdot)$ is the cross-entropy loss used to train the neural network $f$, and $y$ represents the ground truth of $x$. Optimization based methods have also been proposed to optimize adversarial perturbation for targeted attacks while satisfying certain constraints (Carlini & Wagner, 2017a; Liu et al., 2017). Its goal is to minimize the objective function as $||\eta|| + \lambda \ell_f(x_A, y)$, where $||\cdot||$ is an appropriately chosen norm function. However, the optimization process is slow and can only optimize perturbation for one specific instance each time. In contrast, our feed-forward network can produce perturbation for any instance. It achieves higher attack success rate against different defenses and performs much faster than the current attack algorithms.

Independently from our work, feed-forward networks have been applied to generate adversarial perturbation (Baluja & Fischer, 2017). However, Baluja & Fischer combine the re-ranking loss and an $L_2$ norm loss, aiming to constrain the generated adversarial instance to be close to the original one in terms of $L_2$; while we apply a deep neural network as a discriminator to help distinguish the instance with other real images to encourage the perceptual quality of the generated adversarial examples.

**Black-box Attacks** Current learning systems usually do not allow white-box accesses against the model for security reasons. Therefore, there is a great need for black-box attacks analysis. Most of the black-box attack strategies are based on the transferability phenomenon (Papernot et al., 2017), where an adversary can train a local model first and generate adversarial examples against it, hoping the same adversarial examples will also be able to attack the other models. Many learning systems allow query accesses to the model. However, there is little work that can leverage query-based access to target models to construct adversarial samples and move beyond transferability. Papernot

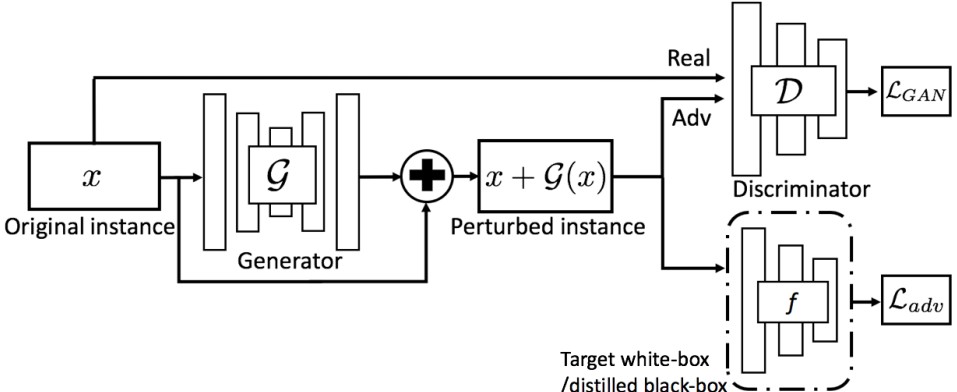

Figure 1: Overview of AdvGAN

et al. (2017) proposed to train a local substitute model with queries to the target model to generate adversarial samples, but this strategy still relies on transferability. In contrast, we show that the proposed AdvGAN can perform black-box attacks without depending on transferability.

**Generative Adversarial Networks (GANs)** (Goodfellow et al., 2014) have achieved visually appealing results in both image generation (Radford et al., 2015; Gulrajani et al., 2017; Berthelot et al., 2017) and manipulation (Zhu et al., 2016) settings. Recently, image-to-image conditional GANs have further improved the quality of synthesis results (Isola et al., 2017; Zhu et al., 2017). We adopt a similar adversarial loss and image-to-image network architecture to learn the mapping from an original image to a perturbed output such that the perturbed image cannot be distinguished from real images in the original class. Different from prior work, we aim to produce output results that are not only visually realistic but also able to mislead target learning models.

## 3 GENERATING ADVERSARIAL EXAMPLES WITH ADVERSARIAL NETWORKS

### 3.1 PROBLEM DEFINITION

Let $\mathcal{X} \subseteq \mathcal{R}^n$ be the feature space, with $n$ the number of features. Suppose that $(x_i, y_i)$ is the $i$th instance within the training set, which is comprised of feature vectors $x_i \in \mathcal{X}$, generated according to some unknown distribution $x_i \sim \mathcal{P}_{\text{data}}$, and $y_i \in \mathcal{Y}$ the corresponding true class labels. The learning system aims to learn a classifier $f : \mathcal{X} \to \mathcal{Y}$ from the domain $\mathcal{X}$ to the set of classification outputs $\mathcal{Y}$, where $|\mathcal{Y}|$ denotes the number of possible classification outputs. Given an instance $x$, the goal of an adversary is to generate adversarial example $x_A$, which is classified as $f(x_A) \neq y$ (untargeted attack), where $y$ denotes the true label; or $f(x_A) = t$ (targeted attack) where $t$ is the target class. $x_A$ should also be close to the original instance $x$ in terms of $L_2$ or other distance metric.

### 3.2 ADVGAN FRAMEWORK

Figure 1 illustrates the overall architecture of AdvGAN, which mainly consists of three parts: a generator $\mathcal{G}$, a discriminator $\mathcal{D}$, and the target neural network $f$. Here the generator $\mathcal{G}$ takes the original instance $x$ as its input and generates a perturbation $\mathcal{G}(x)$. Then $x + \mathcal{G}(x)$ will be sent to the discriminator $\mathcal{D}$, which is used to distinguish the generated data and the original instance $x$. The goal of $\mathcal{D}$ is to encourage that the generated instance is indistinguishable with the data from its original class. To fulfill the goal of fooling a learning model, we first perform the white-box attack, where the target model is $f$ in this case. $f$ takes $x + \mathcal{G}(x)$ as its input and outputs its loss $\mathcal{L}_{adv}$, which represents the distance between the prediction and the target class $t$ (targeted attack), or the opposite of the distance between the prediction and the ground truth class (untargeted attack).

The adversarial loss (Goodfellow et al., 2014) can be written as: [1]

$$\mathcal{L}_{\text{GAN}} = \mathbb{E}_x \log \mathcal{D}(x) + \mathbb{E}_x \log(1 - \mathcal{D}(x + \mathcal{G}(x))). \tag{1}$$

Here, the discriminator $\mathcal{D}$ aims to distinguish the perturbed data $x + \mathcal{G}(x)$ from the original data $x$.[2] Note that the real data is sampled from the true class, so as to encourage that the generated instances are close to data from the original class.

The loss for fooling the target model $f$ in a targeted attack is:

$$\mathcal{L}_{\text{adv}}^f = \mathbb{E}_x \ell_f(x + \mathcal{G}(x), t), \tag{2}$$

where $t$ is the target class and $\ell_f$ denotes the loss function (e.g., cross-entropy loss) used to train the original model $f$. The $\mathcal{L}_{adv}^f$ loss encourages the perturbed image to be misclassified as target class $t$. Here we can also perform the untargeted attack by maximizing the distance between the prediction and the ground truth, but we will focus on the targeted attack in the rest of the paper.

To bound the magnitude of the perturbation, which is a common practice in prior work (Carlini & Wagner, 2017a; Liu et al., 2017; Bartlett & Wegkamp, 2008), we add a soft hinge loss on the $L_2$ norm as

$$\mathcal{L}_{\text{hinge}} = \mathbb{E}_x \max(0, \|\mathcal{G}(x)\|_2 - c), \tag{3}$$

where $c$ denotes a user-specified bound. This can also stabilize the GAN's training, as shown in Isola et al. (2017). Finally, our full objective can be expressed as

$$\mathcal{L} = \mathcal{L}_{\text{adv}}^f + \alpha \mathcal{L}_{\text{GAN}} + \beta \mathcal{L}_{\text{hinge}}, \tag{4}$$

where $\alpha$ and $\beta$ control the relative importance of each objective. Note that $\mathcal{L}_{\text{GAN}}$ here is used to encourage the perturbed data to appear similar to the original data $x$, while $\mathcal{L}_{\text{adv}}^f$ is leveraged to generate adversarial examples, optimizing for the high attack success rate. We obtain our $\mathcal{G}$ and $\mathcal{D}$ by solving the minmax game $\arg \min_{\mathcal{G}} \max_{\mathcal{D}} \mathcal{L}$.

### 3.3 BLACK-BOX ATTACKS WITH ADVERSARIAL NETWORKS

**Static Distillation** For black-box attack, we assume adversaries have no prior knowledge of training data or the model itself. In our experiments in Section 4, we randomly draw data that is *disjoint* from the training data of the black-box model to distill it, since we assume the adversaries have no prior knowledge about the training data or the model. To achieve black-box attacks, we first build a distilled network $f$ based on the output of the black-box model $b$ (Hinton et al., 2015). Once we obtain the distilled network $f$, we carry out the same attack strategy as described in the white-box setting (see Equation (4)). Here, we minimize the following network distillation objective:

$$\arg \min_f \mathbb{E}_x \; \mathcal{H}(f(x), b(x)), \tag{5}$$

where $f(x)$ and $b(x)$ denote the output from the distilled model and black-box model respectively for the given training image $x$, and $\mathcal{H}$ denotes the commonly used cross-entropy loss. By optimizing the objective over all the training images, we can obtain a model $f$ which behaves very close to the black-box model $b$. We then carry out the attack on the distilled network.

Note that unlike training the discriminator $\mathcal{D}$, where we only use the real data from the original class to encourage that the generated instance is close to its original class, here we train the distilled model with data from all classes.

**Dynamic Distillation** Only training the distilled model with all the pristine training data is not enough, since it is unclear how close the black-box and distilled model perform on the generated adversarial examples, which have not appeared in the training set before. Here we propose an *alternative minimization* approach to dynamically make queries and train the distilled model $f$ and our generator $\mathcal{G}$ jointly. We perform the following two steps in each iteration. During iteration $i$:

---

[1] For simplicity, we denote the $\mathbb{E}_x \equiv \mathbb{E}_{x \sim \mathcal{P}_{\text{data}}(x)}$

[2] Note that we only use the generator to produce the perturbation $\mathcal{G}(x)$.

Table 1: Comparison with the state-of-the-art attack methods. Run time is measured for generating 1,000 adversarial instances during test time. Opt. represents the optimization based method, and Trans. denotes black-box attacks based on transferability.

|  | FGSM | Opt. | Trans. | AdvGAN |
|---|---|---|---|---|
| Run time | 0.06s | >3h | - | <0.01s |
| Targeted Attack | ✓ | ✓ | Ens. | ✓ |
| Black-box Attack |  |  | ✓ | ✓ |

1. **Update $\mathcal{G}_i$ given a fixed network** $f_{i-1}$: We follow the white-box setting (see Equation 4) and train the generator and discriminator based on a previously distilled model $f_{i-1}$. We initialize the weights $\mathcal{G}_i$ as $\mathcal{G}_{i-1}$.

$$\mathcal{G}_i, D_i = \arg\min_{\mathcal{G}} \max_{\mathcal{D}} \mathcal{L}_{\text{adv}}^{f_{i-1}} + \alpha\mathcal{L}_{\text{GAN}} + \beta\mathcal{L}_{\text{hinge}} \tag{6}$$

2. **Update $f_i$ given a fixed generator** $\mathcal{G}_i$: First, we use $f_{i-1}$ to initialize $f_i$. Then, given the generated adversarial examples $x + \mathcal{G}_i(x)$ from $\mathcal{G}_i$, the distilled model $f_i$ will be updated based on the set of new query results for the generated adversarial examples against the black-box model, as well as the original training images.

$$f_i = \arg\min_{f} \mathbb{E}_x \mathcal{H}(f(x), b(x)) + \mathbb{E}_x \mathcal{H}(f(x + \mathcal{G}_i(x)), b(x + \mathcal{G}_i(x))), \tag{7}$$

where we use both the original images $x$ and the newly generated adversarial examples $x + s\mathcal{G}_i(x)$ to update $f$.

In the experiment section, we compare the performance of both the static and dynamic distillation approaches and observe that simultaneously updating $\mathcal{G}$ and $f$ produces higher attack performance. See Table 2 for more details.

## 4 EXPERIMENTAL RESULTS

In this section, we first evaluate AdvGAN for both semi-whitebox and black-box settings on MNIST (LeCun & Cortes, 1998) and CIFAR-10 (Krizhevsky et al., 2014). We also perform a semi-whitebox attack on the ImageNet dataset(Deng et al., 2009). We then apply AdvGAN to generate adversarial examples on different target models and test the attack success rate for them under the state-of-the-art defenses and show that our method can achieve higher attack success rates compared to other existing attack strategies. We generate all adversarial examples for different attack methods based on the under $L_\infty$ bound of 0.3 on MNIST and 8 on CIFAR-10, for a fair comparison.

In general, as shown in Table 1, AdvGAN has several advantages over other white-box and black-box attacks. For instance, regarding computation efficiency, AdvGAN performs much faster than others even including the efficient FGSM, although AdvGAN needs extra training time to train the generator. All these strategies can perform targeted attack except transferability based attack, although the ensemble strategy can help to improve. Besides, FGSM and optimization methods can only perform white-box attack, while AdvGAN is able to attack in semi-whitebox setting.

**Implementation Details** Our code and models will be available upon publication. We adopt a similar architecture from image-to-image translation literature (Isola et al., 2017; Zhu et al., 2017). In particular, we use the architecture of generator $\mathcal{G}$ from Johnson et al. (2016), and our discriminator $\mathcal{D}$'s architecture is similar to model C for MNIST and ResNet-32 for CIFAR-10. We apply the loss in Carlini & Wagner (2017c) as our loss $\mathcal{L}_{adv}^f = \max(\max_{i \neq t} f(x_A)_i - f(x_A)_t, \kappa)$, where $t$ is the target class, and $f$ represents the target network in the semi-whitebox setting and the distilled model in the black-box setting. We set the confidence $\kappa = 0$ for both Opt. and AdvGAN. We use Adam as our solver (Kingma & Ba, 2014), with a batch size of 128 and a learning rate of 0.001. For GANs training, we use the least squares objective proposed by LSGAN (Mao et al., 2016), as it has been shown to produce better results with more stable training.

Table 2: Accuracy of different models on pristine data, and the attack success rate of adversarial examples generated against different models by AdvGAN on MNIST and CIFAR-10. p: pristine test data; w: semi-whitebox attack; b-D: black-box attack with dynamic distillation strategy; b-S: black-box attack with static distillation strategy.

| Model | MNIST | | | CIFAR-10 | |
|---|---|---|---|---|---|
| | A | B | C | ResNet-32 | Wide ResNet-34 |
| Accuracy (p) | 98.97% | 99.17% | 99.09% | 92.41% | 95.01% |
| Attack Success Rate (w) | 97.9% | 97.1% | 98.3% | 94.71% | 99.30% |
| Attack Success Rate (b-D) | 93.4% | 90.1% | 94.02% | 78.47 % | 81.81% |
| Attack Success Rate (b-S) | 30.7% | 66.63% | 87.3% | 10.3% | 13.3% |

**Models Used in the Experiments**  For MNIST, in all of our experiments, we generate adversarial examples for three models whose architectures are shown in Appendix A. Models A and B are used in Tramèr et al. (2017b), which represent different architectures. Model C is the target network architecture used in (Carlini & Wagner, 2017a) for evaluating optimization based strategy. For CIFAR-10, we select ResNet-32 and Wide ResNet-34 (He et al., 2016; Zagoruyko & Komodakis, 2016) for our experiments. Specifically, we use a 32-layer ResNet implemented in TensorFlow[3] and Wide ResNet derived from the variant of "w32-10 wide."[4] We show the classification accuracy of pristine MNIST and CIFAR-10 test data (p) and attack success rate of adversarial examples generated by AdvGAN on different models in Table 2.

## 4.1 ADVGAN IN SEMI-WHITEBOX SETTING

First, we apply different architectures for the target model $f$ as listed in Appendix A for MNIST and with ResNet and Wide ResNet for CIFAR-10. We first apply AdvGAN to perform semi-whitebox attack against each model on MNIST dataset. From the performance of semi-whitebox attack (Attack Rate (w)) in Table 2, we can see that AdvGAN is able to generate adversarial instances to attack all models with high attack success rate.

We also generate adversarial examples from the same original instance $x$, targeting other different classes, as shown in Figures 2. In the semi-whitebox setting on MNIST (a)-(c), we can see that the generated adversarial examples for different models appear close to the ground truth/pristine images (lying on the diagonal of the matrix). Figure 2 (d)-(f) show the generated adversarial examples on MNIST in black-box setting. These adversarial examples generated by AdvGAN can successfully fool the black-box model and be misclassified as the target class shown on the top. The original images are shown on the diagonal. We also generate adversarial examples based on random original images, and results are shown in Appendix C.

In addition, we analyze the attack success rate based on different loss functions on MNIST. Under the same bounded perturbations (0.3), if we replace the full loss function in (4) with $\mathcal{L} = ||\mathcal{G}(x)||_2 + \mathcal{L}_{\text{adv}}^f$, which is similar to the objective used in Baluja & Fischer (2017), the attack success rate becomes 86.2%. If we replace the loss function with $\mathcal{L} = \mathcal{L}_{\text{hinge}} + \mathcal{L}_{\text{adv}}^f$, the attack success rate is 91.1%, compared to that of AdvGAN, 98.3%.

Similarly, on CIFAR-10, we apply the same semi-whitebox attack for ResNet and Wide ResNet based on AdvGAN, and Figure 3(a) shows some adversarial examples, which are perceptually realistic. We show adversarial examples for the same original instance targeting different other classes. It is clear that with different targets, the adversarial examples keep similar visual quality compared to the pristine instances on the diagonal.

We also apply AdvGAN to generate adversarial examples on the ImageNet as shown in Figure 4 with $L_\infty$ bound as 8. The added perturbation is unnoticeable while all the adversarial instances are misclassified into other target classes with high confidence.

---

[3]https://github.com/tensorflow/models/blob/master/research/ResNet/ResNet_model.py
[4]https://github.com/MadryLab/cifar10_challenge/blob/master/model.py

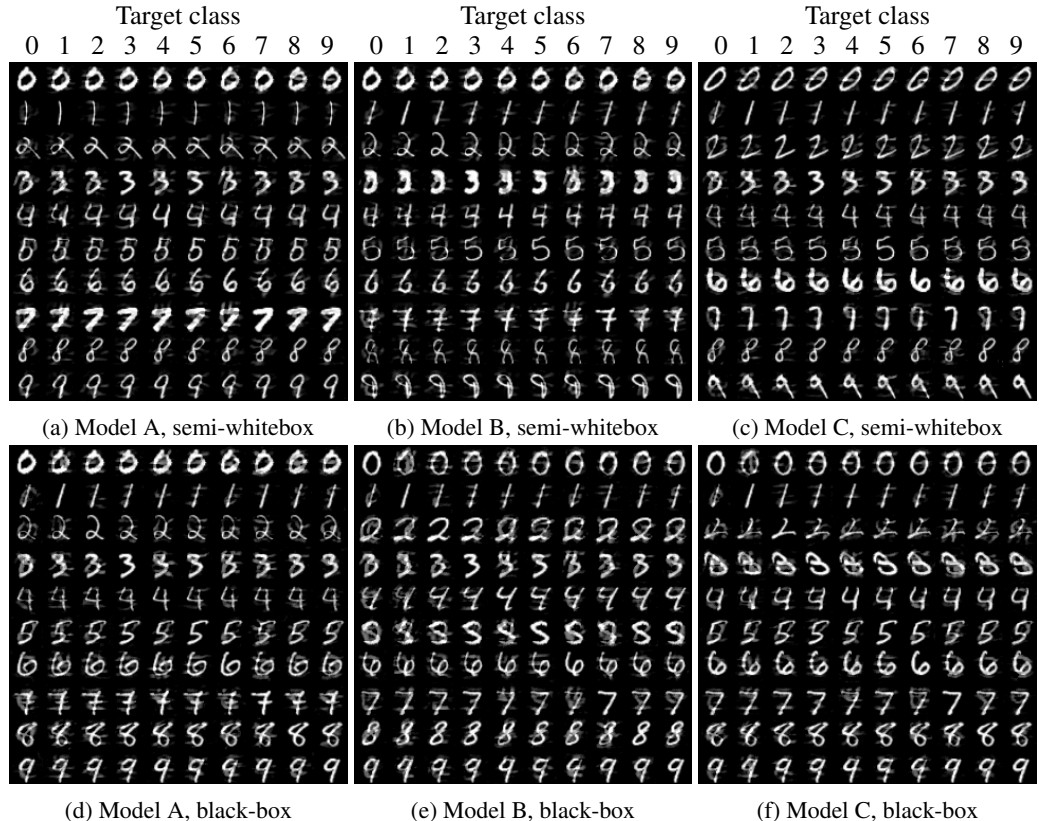

Figure 2: Adversarial examples generated from the same original image to different targets by Adv-GAN on MNIST with semi-whitebox attack, (a), (b), and (c), and black-box attack, (c), (d), and (e). On the diagonal, the original images are shown.

## 4.2 ADVGAN IN BLACK-BOX SETTING

In this section, we evaluate the performance of AdvGAN for the black-box attack. Our black-box attack here is based on the dynamic distillation strategy. We construct a local model to distill model $f$, and we select the architecture of Model C as our local model. Note that we randomly select a subset of instances disjoint from the training data of AdvGAN to train the local model; that is, we assume the adversaries do not have any prior knowledge of the training data or the model itself. With the dynamic distillation strategy, the adversarial examples generated by AdvGAN achieve an attack success rate, above $90\%$ for MNIST and $80\%$ for CIFAR-10, compared to $30\%$ and $10\%$ with the static distillation approach, as shown in Table 2.

We apply AdvGAN to generate adversarial examples for the same instance targeting different classes on MNIST and randomly select some instances to show in Figure 2 (d)-(f). By comparing with the pristine instances on the diagonal, we can see that these adversarial instances can achieve high perceptual quality as the original digits. Specifically, the original digit is somewhat highlighted by adversarial perturbations, which implies a type of perceptually realistic manipulation. Figure 3 (b) shows similar results for adversarial examples generated on CIFAR-10. These adversarial instances appear photo-realistic compared with the original ones on the diagonal. We show additional results in Appendix C.

## 4.3 ATTACK EFFECTIVENESS UNDER DEFENSES

Facing different types of attack strategies, various defenses have been provided. Among them, different types of adversarial training methods are the most effective. Goodfellow et al. (2015) first propose adversarial training as an effective way to improve the robustness of DNNs, and Tramèr

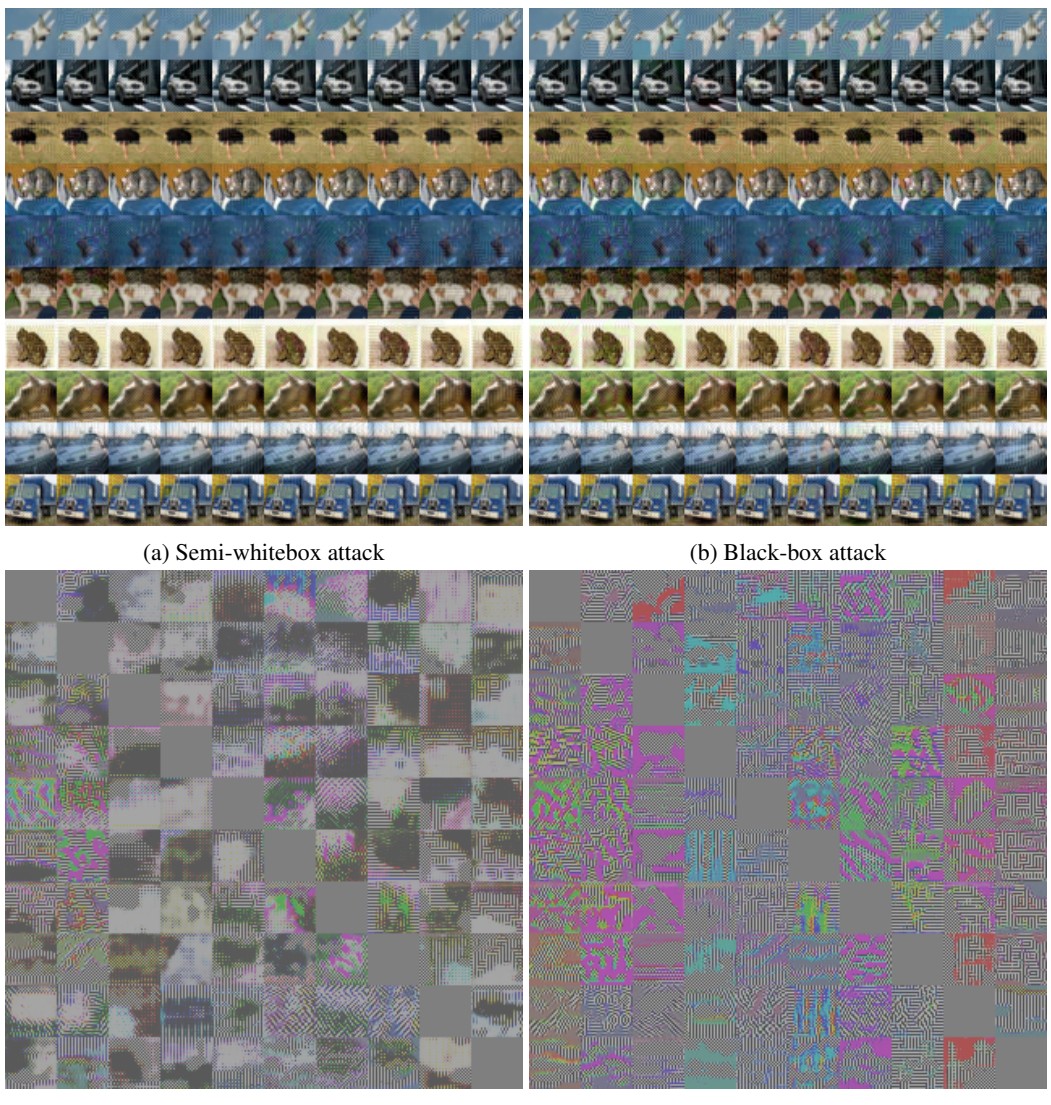

(a) Semi-whitebox attack          (b) Black-box attack

(c) Perturbations generated in semi-whitebox attack     (d) Perturbations generated in black-box attack

Figure 3: Adversarial examples generated by AdvGAN on CIFAR-10 for (a) semi-whitebox attack and (b) black-box attack. Image from each class is perturbed to other different classes. On the diagonal, the original images are shown. The corresponding perturbations (amplified by $10\times$) are shown in (c) and (d).

et al. (2017a) extend it to ensemble adversarial learning. Mądry et al. (2017a) have also proposed robust networks against adversarial examples based on well-defined adversaries. Given the fact that AdvGAN strives to generate adversarial instances from the underlying true data distribution, it can essentially produce more photo-realistic adversarial perturbations compared with other attack strategies. Thus, AdvGAN could have a higher chance to produce adversarial examples that are resilient under different defense methods. In this section, we quantitatively evaluate this property for AdvGAN compared with other attack strategies.

**Threat Model** As shown in the literature, most of the current defense strategies are not robust when attacking against them (Carlini & Wagner, 2017b; He et al., 2017). Here we consider a weaker threat model, where the adversary is not aware of the defenses and directly tries to attack the original learning model, which is also the first threat model analyzed in Carlini & Wagner (2017b). In this case, if an adversary can still successfully attack the model, it implies the robustness of the attack strategy. Under this setting, we first apply different attack methods to generate adversarial examples

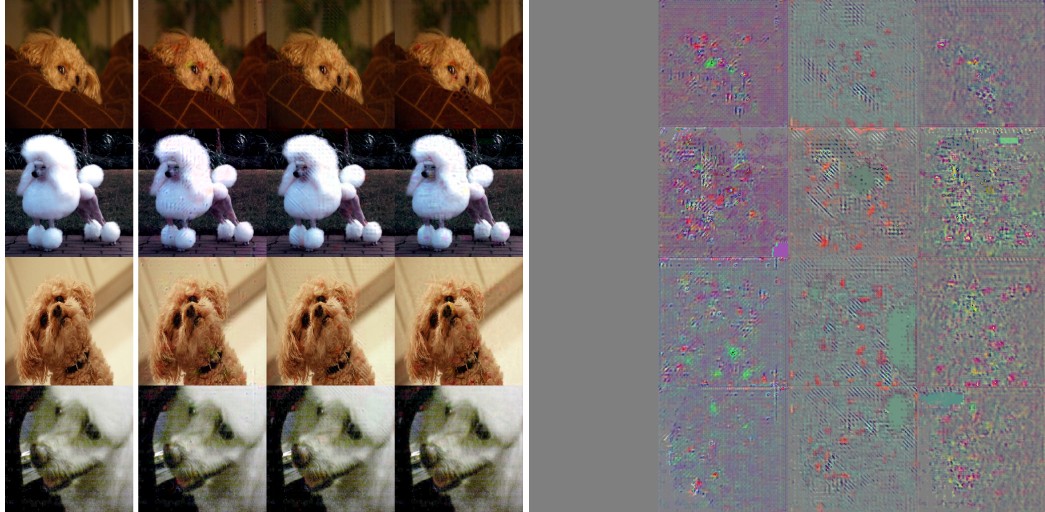

(a) Adversarial examples         (b) Corresponding perturbations (amplified by $10\times$)

Figure 4: Adversarial examples (a) generated by AdvGAN on ImageNet in the semi-whitebox setting, which are classified as (from left to right) poodle, ambulance, basketball, and electric guitar. Corresponding perturbations are visualized in (b).

based on the original model without being aware of any defense. Then we apply different defenses to directly defend against these adversarial instances.

**Semi-whitebox Attack**  First, we consider the semi-whitebox attack setting, where the adversary has white-box access to the model architecture as well as the parameters. Here, we replace $f$ in Figure 1 with our model A, B, and C, respectively. As a result, adversarial examples will be generated against different models. We use three adversarial training defenses to train different models for each model architecture: standard FGSM adversarial training (Adv.) (Goodfellow et al., 2015), ensemble adversarial training (Ensemble) (Tramèr et al., 2017b), and iterative training (Iter. Adv.) (Mądry et al., 2017a).[5] We evaluate the effectiveness of these attacks against these defended models. In Table 3, we show that the attack success rate of adversarial examples generated by AdvGAN on different models is higher than those of the fast gradient sign method (FGSM) and optimization methods (Opt.) (Carlini & Wagner, 2017a).

**Black-box Attack**  For AdvGAN, we use model B as the black-box model and train a distilled model to perform black-box attack against model B and report the attack success rate in Table 4. For the black-box attack comparison purpose, transferability based attack is applied for FGSM and optimization-based methods (Opt.). We use FGSM and optimization-based methods (Opt.) to attack model A on MNIST, and we use these adversarial examples to test on model B and report the corresponding classification accuracy. We can see that the adversarial examples generated by the black-box AdvGAN consistently achieve much higher attack success rate compared with other attack methods. For CIFAR-10, we use ResNet as black-box model and train a distilled model to perform black-box attack against ResNet. To evaluate black-box attack for optimization method and FGSM, we use adversarial examples generated by attacking Wide ResNet and test them on ResNet to report black-box attack results for these two methods.

In addition, we apply AdvGAN to the MNIST challenge (Mądry et al., 2017b). Among all the methods, for white-box attack we achieve 88.93% accuracy on the published local model as shown in Table 5. For the reported black-box attack, we achieved the accuracy as 92.76%, outperforming all other state-of-the-art attack strategies.

---

[5] Each ensemble adversarially trained model is trained using (i) pristine training data, (ii) FGSM adversarial examples generated for the current model under training, and (iii) FGSM adversarial examples generated for naturally trained models of two architectures different from the model under training.

Table 3: Attack success rate of adversarial examples generated by AdvGAN in semi-whitebox setting, and other white-box attacks under defenses on MNIST and CIFAR-10.

| Data | Model | Defense | FGSM | Opt. | AdvGAN |
|---|---|---|---|---|---|
| MNIST | A | Adv. | 4.3% | 4.6% | **8.0%** |
| | | Ensemble | 1.6% | 4.2% | **6.3%** |
| | | Iter.Adv. | 4.4% | 2.96% | **5.6%** |
| | B | Adv. | 6.0% | 4.5% | **7.2%** |
| | | Ensemble | 2.7% | 3.18% | **5.8%** |
| | | Iter.Adv. | **9.0%** | 3.0% | 6.6% |
| | C | Adv. | 2.7% | 2.95% | **18.7%** |
| | | Ensemble | 1.6% | 2.2% | **13.5%** |
| | | Iter.Adv. | 1.6% | 1.9% | **12.6%** |
| CIFAR | ResNet | Adv. | 13.10% | 11.9% | **16.03%** |
| | | Ensemble. | 10.00% | 10.3% | **14.32%** |
| | | Iter.Adv | 22.8% | 21.4% | **29.47%** |
| | Wide ResNet | Adv. | 5.04% | 7.61% | **14.26%** |
| | | Ensemble | 4.65% | 8.43% | **13.94 %** |
| | | Iter.Adv. | 14.9% | 13.90% | **20.75%** |

Table 4: Attack success rate of adversarial examples generated by different black-box adversarial strategies under defenses on MNIST and CIFAR-10

| | MNIST | | | CIFAR-10 | | |
|---|---|---|---|---|---|---|
| Defense | FGSM | Opt. | AdvGAN | FGSM | Opt. | AdvGAN |
| Adv. | 3.1% | 3.5% | **11.5%** | 13.58% | 10.8% | **15.96%** |
| Ensemble | 2.5% | 3.4% | **10.3%** | 10.49% | 9.6% | **12.47%** |
| Iterative Adv. | 2.4% | 2.5% | **12.2%** | 22.96% | 21.70% | **24.28%** |

Table 5: Accuracy of the MadryLab public model under different attacks in white-box setting. The AdvGAN here achieved the best performance.

| Method | Accuracy (xent loss) | Accuracy (cw loss) |
|---|---|---|
| FGSM | 95.23% | 96.29% |
| PGD | 93.66% | 93.79% |
| Opt | - | 91.69% |
| **AdvGAN** | - | **88.93%** |

## 4.4 ADVERSARIAL PERTURBATION ANALYSIS.

To understand the adversarial perturbation pattern better, we plot out corresponding perturbations (amplified by a factor of 10) for CIFAR-10 in Figure 3 (c) and (d) and ImageNet in Figure 4 (b). From the visualization of perturbation, it shows that the perturbations do not resemble anything in particular about the original image or the target class. Although training AdvGAN exposes it to realistic instances, the perturbations it generates do not simply interpolate towards an example of the target class.

## 4.5 HIGH RESOLUTION ADVERSARIAL EXAMPLES ANALYSIS

To evaluate AdvGAN's ability to generate high resolution adversarial examples, we generate the high resolution adversarial examples for Inception_v3 and quantify their attack success rate and perceptual realism.

**Experiment settings.** In the following experiments, we select *toy poodle* as our target label for all images. We select 100 benign images from the DEV set of the NIPS 2017 targeted adversarial

attack competition.[6] This competition provided a dataset compatible with ImageNet. We generate adversarial examples (299×299 pixels) under an $L_\infty$ perturbation bound of 0.01 (pixels values are in the range $\in [0,1]$) for the Inception_v3 model, whose input size is 299×299. The details of architectures for generator and discriminator we used are listed in Appendix D.

Table 6: Parameters of generated high resolution adversarial examples

| Dateset | Model | Target Label | Resolution | $L_\infty$ bound | Attack Success Rate |
|---|---|---|---|---|---|
| ImageNet | Inception_v3 | toy poodle | $299 \times 299$ | 0.01 | 100% |

In Figure 8 in the appendix, we show the original images on the left with the correct label, and we show adversarial examples generated by AdvGAN on the right with the target label.

**Human Perceptual Study.** We validate the realism of AdvGAN's adversarial examples with a user study on Amazon Mechanical Turk (AMT). We use 100 pairs of original images and adversarial examples (generated as described above) and ask workers to choose which image of a pair is more *visually realistic*.

Our study follows a protocol from Zhang et al. (2016) and Isola et al. (2017), where a worker is shown a pair of images for 2 seconds, then the worker has unlimited time to choose. We limit each worker to at most 20 of these tasks. We collected 500 choices, about 5 per pair of images, from 50 workers on AMT.

The AdvGAN examples were chosen as more realistic than the original image in $49.4\% \pm 1.96\%$ of the tasks (random guessing would result in about $50\%$). This result show that these high-resolution AdvGAN adversarial examples are about as realistic as benign images.

## 5 CONCLUSION

In this paper, we propose AdvGAN to generate adversarial examples using generative adversarial networks (GANs). In our AdvGAN framework, once trained, the feed-forward generator can produce adversarial perturbations efficiently. It can also perform both semi-whitebox and black-box attacks with high attack success rate. In addition, when we apply AdvGAN to generate adversarial instances on different models without knowledge of the defenses in place, the generated adversarial examples can attack the state-of-the-art defenses with higher attack success rate than examples generated by the competing methods. This property makes AdvGAN a promising candidate for improving adversarial training defense methods. The generated adversarial examples produced by AdvGAN preserve high perceptual quality due to GANs' distribution approximation property.

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

## A ARCHITECTURE OF MODELS

Table 7: Model architectures for the MNIST

| A | B | C |
|---|---|---|
| Conv(64,5,5)+Relu | Conv(64,8,8)+Relu | Conv(32,3,3)+Relu |
| Conv(64,5,5)+Relu | Dropout(0.2) | Conv(32,3,3)+Relu |
| Dropout(0.25) | Conv(128, 6, 6)+Relu | MaxPooling(2,2) |
| FC(128)+Relu | Conv(128, 5, 5)+Relu | Conv(64,3,3)+Relu |
| Dropout(0.5) | Dropout(0.5) | Conv(64,3,3)+Relu |
| FC(10)+Softmax | FC(10)+Softma | MaxPooling(2,2) |
| | | FC(200)+Relu |
| | | FC(200)+Softmax |

## B NETWORK ARCHITECTURES

**Generator architecture** We follow the naming rules used in Johnson et al. (2016)'s Github repository[7] as well as Zhu et al. (2017) . Let c3s1-k denotes $3 \times 3$ Convolution-InstanceNorm-ReLU layer with k filter and stride 1. Rk means residual block that contains two $3 \times 3$ convolution layers with the same numbers of filters. dk denotes the $3 \times 3$ Convolution-InstanceNorm-ReLU layer with k filters and stride 2. uk denotes a $3 \times 3$ fractional-strided-ConvolutionInstanceNorm-ReLU layer with k filters, and stride $\frac{1}{2}$. .

The generator structures consists of:
c3s1-8, d16, d32, r32, r32, r32, r32, u16, u8, c3s1-3

**Discriminator architecture** We use CNNs as our discriminator network (Radford et al., 2015). Let Ck denote a $4 \times 4$ Convolution-InstanceNorm-LeakyReLU layer with k filters and stride 2. After the last conv layer, we apply a FC layer to produce a 1 dimensional output. We do not use InstanceNorm for the first C8 layer. We use leaky ReLUs with slope 0.2.

The discriminator architecture is:
C8, C16, C32, FC

## C ADDITIONAL ADVERSARIAL EXAMPLES

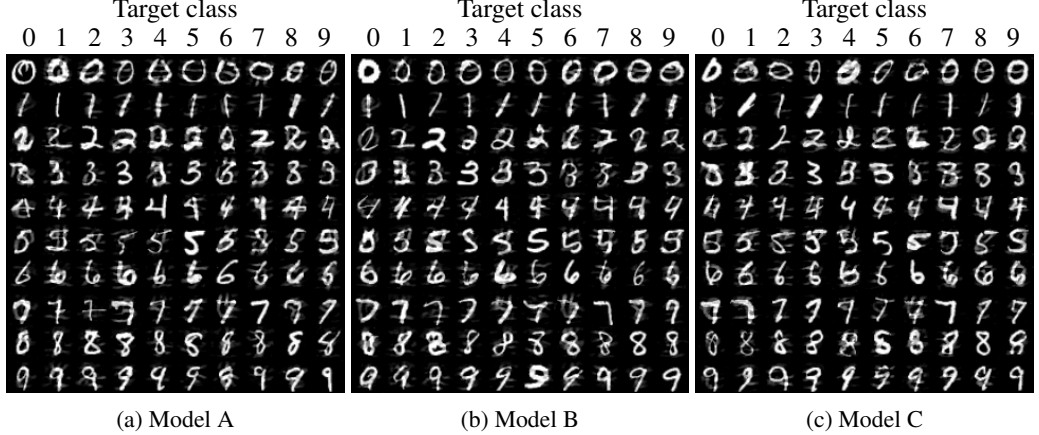

(a) Model A          (b) Model B          (c) Model C

Figure 5: Adversarial examples generated by AdvGAN on MNIST against different models in the semi-whitebox setting. Here the adversarial examples are randomly sampled corresponding to different original images.

---

[7]https://github.com/jcjohnson/fast-neural-style.

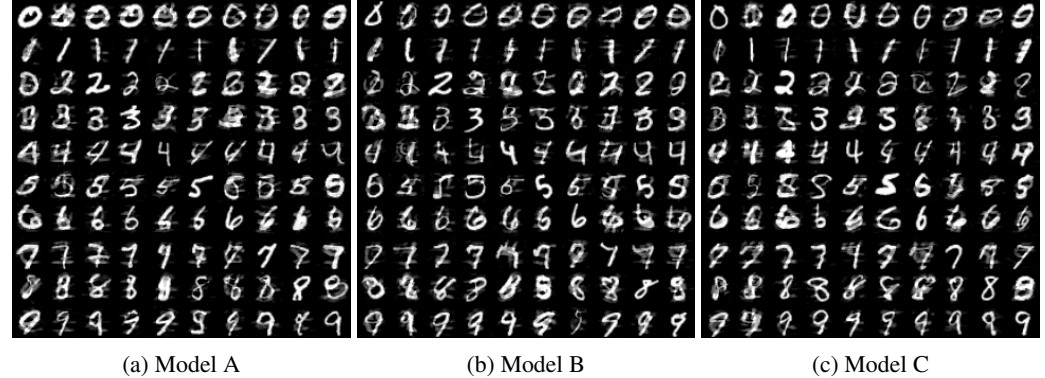

| (a) Model A | (b) Model B | (c) Model C |

Figure 6: Adversarial examples generated by AdvGAN on MNIST against different models in the black-box setting. Here the adversarial examples are randomly sampled corresponding to different original images.

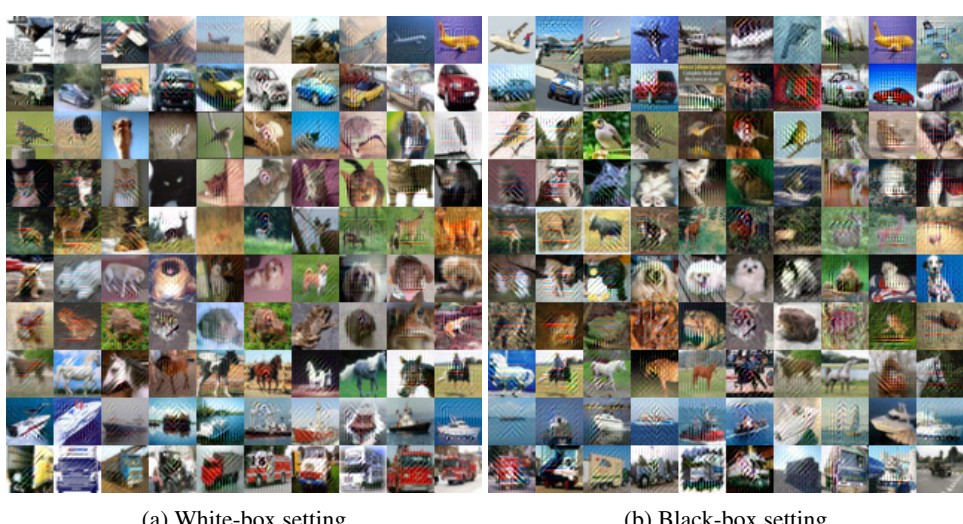

| (a) White-box setting | (b) Black-box setting |

Figure 7: Adversarial examples generated by AdvGAN on CIFAR-10. Here the adversarial examples are randomly sampled corresponding to different original images.

Table 8: Comparisons of perturbations generated by AdvGAN and the state-of-the-art algorithms on MNIST and CIFAR-10. We report the mean value of perturbation amount as "mean" and attack success rate as "prob."

| Method | MNIST | | | | | | CIFAR-10 | | | |
| | A | | B | | C | | ResNet-32 | | Wide ResNet-34 | |
| | mean | prob | mean | prob | mean | prob | mean | prob | mean | prob |
| AdvGAN | 0.149 | 98% | 0.157 | 97% | 0.144 | 98% | 0.025 | 95% | 0.024 | 99% |
| CW | 0.089 | 99% | 0.100 | 99% | 0.070 | 100% | 0.023 | 100% | 0.020 | 98% |
| FGSM | 0.202 | 55% | 0.193 | 49% | 0.192 | 18% | 0.0301 | 23% | 0.031 | 26% |

# D HIGH RESOLUTION ADVERSARIAL EXAMPLES FOR AN IMAGENET-COMPATIBLE SET

The structure of generator for ImageNet consists of:

```
c7s1-8, d16, d32, d64, d64, d64, d64, r64, r64, r64, r64, u64,
u64, u64, u64, u32, u16, u8, c7s1-3
```

The architecture of discriminator for ImageNet is:
```
C8, C16, C32, FC
```

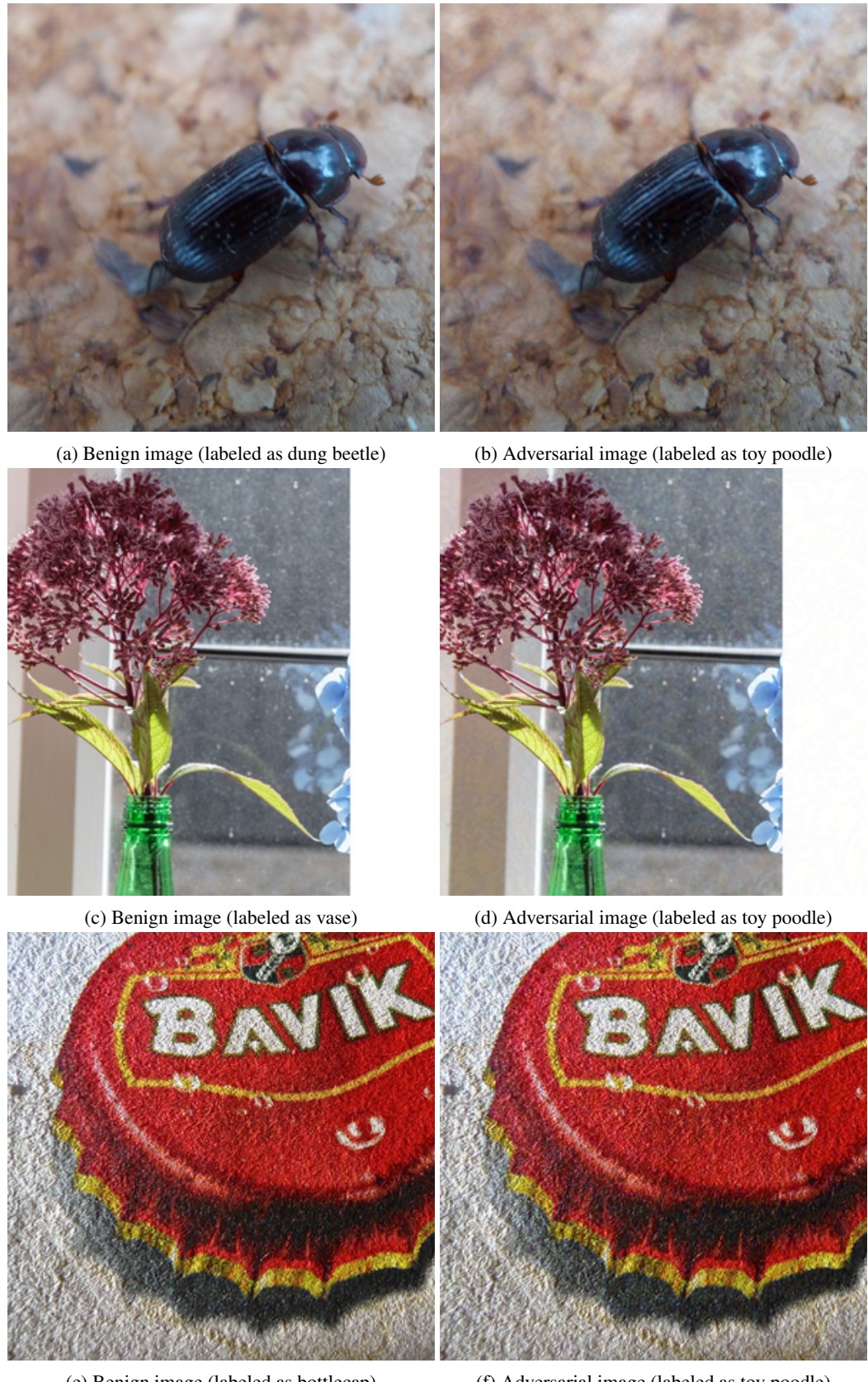

(a) Benign image (labeled as dung beetle)     (b) Adversarial image (labeled as toy poodle)

(c) Benign image (labeled as vase)     (d) Adversarial image (labeled as toy poodle)

(e) Benign image (labeled as bottlecap)     (f) Adversarial image (labeled as toy poodle)

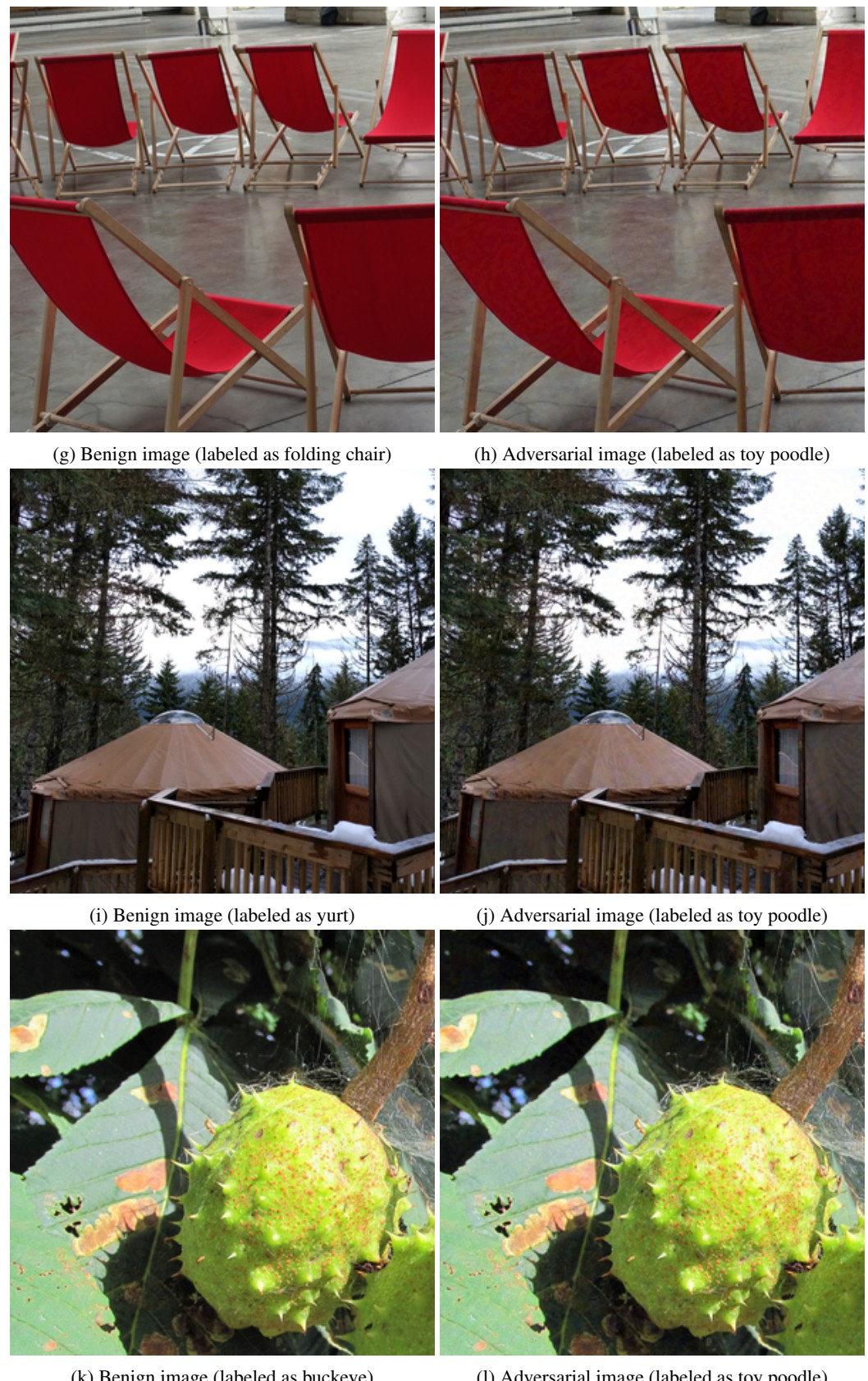

(g) Benign image (labeled as folding chair)        (h) Adversarial image (labeled as toy poodle)

(i) Benign image (labeled as yurt)        (j) Adversarial image (labeled as toy poodle)

(k) Benign image (labeled as buckeye)        (l) Adversarial image (labeled as toy poodle)

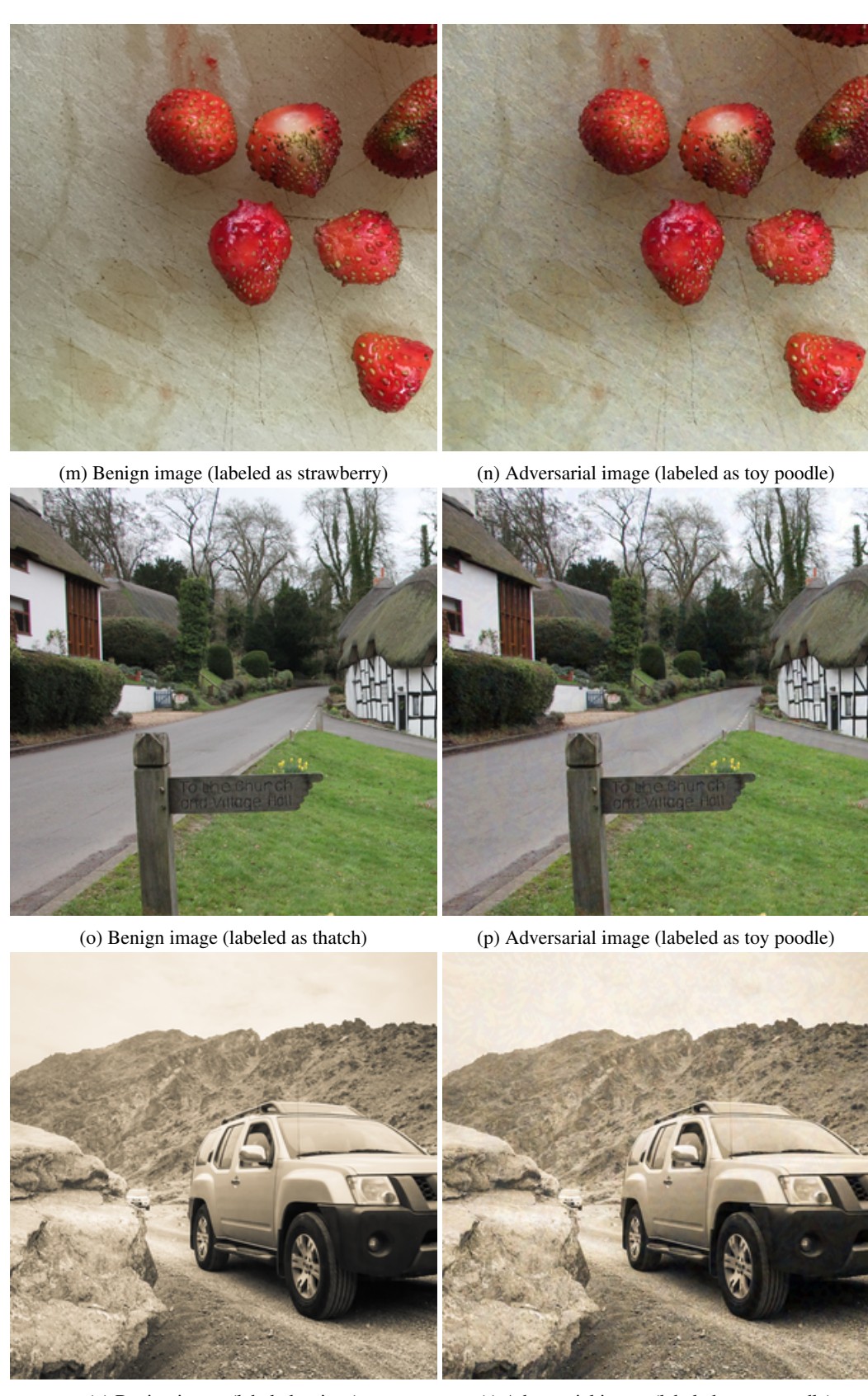

(m) Benign image (labeled as strawberry)        (n) Adversarial image (labeled as toy poodle)

(o) Benign image (labeled as thatch)        (p) Adversarial image (labeled as toy poodle)

(q) Benign image (labeled as jeep)        (r) Adversarial image (labeled as toy poodle)

Figure 8: Examples from an ImageNet-compatible set. Left: original image; right: adversarial image generated by AdvGAN against Inception_v3.

