# OpenReview forum: "Generating Adversarial Examples with Adversarial Networks"
_ICLR.cc/2018/Conference — Reject_

### Official Review · AnonReviewer2 · 2017-11-17
**Review for 'Generating Adversarial Examples with Adversarial Networks'**

**Rating:** 4
**Confidence:** 4

**Review:**

I thank the authors for the thoughtful response and rebuttal. The authors have substantially updated their manuscript and improved the presentation.

Re: Speed. I brought up this point because this was a bulleted item in the Introduction in the earlier version of the manuscript. In the revised manuscript, this bullet point is now removed. I will take this point to be moot.

Re: High resolution. The authors point to recent GAN literature that provides some first results with high resolution GANs but I do not see quantitative evidence in the high resolution setting for this paper. (Figure 4 provides qualitative examples from ImageNet but no quantitative assessment.)

Because the authors improved the manuscript, I upwardly revised my score to 'Ok but not good enough - rejection'. I am not able to accept this paper because of the latter point.
==========================

The authors present an interesting new method for generating adversarial examples. Namely, the author train a generative adversarial network (GAN) to adversarial examples for a target network. The authors demonstrate that the network works well in the semi-white box and black box settings.

The authors wrote a clear paper with great references and clear descriptions.

My primary concern is that this work has limited practical benefit in a realistic setting. Addressing each and every concern is quite important:

1) Speed. The authors suggest that training a GAN provides a speed benefit with respect to other attack techniques. The FGSM method (Goodfellow et al, 2015) is basically 1 inference operation and 1 backward operation. The GAN is 1 forward operation. Granted this results in a small difference in timing 0.06s versus 0.01s, however it would seem that avoiding a backward pass is a somewhat small speed gain.

Furthermore, I would want to question the practical usage of having an 'even faster' method for generating adversarial examples. What is the reason that we need to run adversarial attacks 'even faster'? I am not aware of any use-cases, but if there are some, the authors should describe the rationales at length in their paper.

2) High spatial resolution images. Previous methods, e.g. FGSM, may work on arbitrarily sized images. At best, GANs generate reasonable images that are lower resolutions (e.g. < 128x128). Building GAN's that operate above-and-beyond moderate spatial resolution is an open research topic. The best GAN models for generating high resolution images are  difficult to train and it is not clear if they would work in this setting. Furthermore, images with even higher resolutions, e.g. 512x512, which is quite common in ImageNet, are difficult to synthesizes using current techniques.

3) Controlling the amount of distortion. A feature of previous optimization based methods is that a user may specify the amount of perturbation (epsilon). This is a key feature if not requirement in an adversarial perturbation because a user might want to examine the performance of a given model as a function of epsilon. Performing such an analysis with this model is challenging (i.e. retraining a GAN) and it is not clear if a given image generated by a GAN will always achieve a given epsilon perturbation/

On a more minor note, the authors suggest that generating a *diversity* of adversarial images is of practical import. I do not see the utility of being able to generate a diversity of adversarial images. The authors need to provide more justification for this motivation.

---

> ### Author Response · Authors · 2018-01-04
> **Reply to review**
>
> We thank the reviewer for the thoughtful comments and suggestions.
>
> Speed. Speed is just one advantage of our method and is not the main motivation of our method. We agree with the reviewer that FGSM is already fast enough for most applications. However, our proposed model is much more effective in the fooling rate against both white-box and blackbox settings than the FGSM method and is still faster than the FGSM method. In wide resnet setting on CIFAR, FGSM takes 0.39s to generate 100 examples (1 forward and 1 backward pass through the classifier), while AdvGAN takes 0.16s to generate 100 examples (1 forward pass through the generator). In our experiments, the classifier was a wide resnet with 46.16M parameters, while the generator had 0.24M parameters. The speed difference is even larger with deeper classifiers. Moreover, in AdvGAN, the forward operation does not use classifier’s network, but uses generator’s network. Overall, we claim to have developed a faster and more effective alternative method to generating adversarial examples, but improving the speed is just a byproduct for us and generating more photorealistic and effective adversarial examples in both semi white-box and blackbox settings is the main goal.
>
> High spatial resolution images. Early GANs have had this problem. However, we claim that AdvGAN still works at high spatial resolution (and it is not unique in doing so). Here are three techniques we applied in AdvGAN.
> (i) Previous work in high resolution: Our method is built on image-to-image translation and conditional GANs  (e.g. [pix2pix], [CycleGAN]) rather than unconditional GANs (e.g. [vanilla GANs], [DCGAN]). Many conditional GAN methods have been shown to be able to produce photorealistic results at relatively high resolution (256x256 and 512x512 from [pix2pix] and [CycleGAN]).  The recent pix2pixHD paper on arXiv from NVIDIA [Wang et al. 2017] can even produce 2k photo-realistic images. Even recent unconditional GANs like progressive GANs [Karras et al. 2017] are able to produce 1k images.
> (ii) Retaining details from original image: Our goal is to produce the perturbation rather than the final image: output = input + G(input). Details and textures are copied from the input image.
> (iii) Resolution-independent architecture: Our model is fully convolutional and can be applied to input images with arbitrary sizes, similar to [pix2pix] and [CycleGAN].
>
> Controlling the amount of distortion. We added more detailed description in the updated paper about how we control the amount of perturbation. Basically, we use parameter c within the hinge loss as shown in eq. (3) to allow users to specify the perturbation amount (epsilon). Note that AdvGAN can explicitly control the amount of perturbation since in the MNIST challenge, it is strictly required that the perturbation is bounded within 0.3 in terms of L-infinity. So the competition results also show that we are able to bound the perturbation accurately so as to win the challenge.
>
> Why are we interested in the diversity of adversarial examples? We have seen that ensemble adversarial training works better than adversarial training against FGS + rand [Florian et al. 2017]. This indicates that more diverse adversarial examples are needed to perform adversarial training as a defense. In addition, exploring other diverse adversarial examples can help us better understand the space of adversarial examples. For these reasons, we are interested in how to produce diverse adversarial examples, but indeed, we have not made it the main goal of AdvGAN.
>
> Reference
> [Pix2pix] Isola, Phillip, et al. "Image-to-image translation with conditional adversarial networks." arXiv preprint arXiv:1611.07004 (2016).
> [CycleGAN] Zhu, Jun-Yan, et al. "Unpaired image-to-image translation using cycle-consistent adversarial networks." arXiv preprint arXiv:1703.10593 (2017).
> [Vanilla GAN] Goodfellow, Ian, et al. "Generative adversarial nets." Advances in neural information processing systems. 2014.
> [DCGAN] Radford, Alec, Luke Metz, and Soumith Chintala. "Unsupervised representation learning with deep convolutional generative adversarial networks." arXiv preprint arXiv:1511.06434 (2015).
> [Wang et al. 2017] Wang, Ting-Chun, et al. "High-Resolution Image Synthesis and Semantic Manipulation with Conditional GANs." arXiv preprint arXiv:1711.11585 (2017).
> [Karras et al. 2017] Karras, Tero, et al. "Progressive growing of gans for improved quality, stability, and variation." arXiv preprint arXiv:1710.10196 (2017).
> [Florian et al. 2017] Tramèr, Florian, et al. "Ensemble Adversarial Training: Attacks and Defenses." arXiv preprint arXiv:1705.07204 (2017).

---

> ### Author Response · Authors · 2018-01-11
> **reply of "high resolution"**
>
> Thanks for the valuable feedback. Regarding the reviewer’s request about high resolution images, we have added (1) a quantitative experiment on attack success rates, (2) a user study on perceptual realism of the examples, and (3) additional qualitative examples, which demonstrate that AdvGAN can effectively generate high resolution adversarial examples. The details are as follows.
>
> We generate 100 high resolution (299x299) adversarial examples under an L_infinity bound of 0.01 (pixel values are in range [0,1]). This competition provided a dataset compatible with ImageNet. We observe that the attack success rate of AdvGAN is 100%. Section 4.5 details the experiment settings.
>
> In order to evaluate the perceptual realism of high resolution adversarial examples generated by AdvGAN,  we have added a human study in Section 4.5. In our study, participants chose AdvGAN’s adversarial examples as more realistic over the original images in 49.4% of the trials (matching the realism of the original images results in around 50%). This experiment shows that these high resolution AdvGAN adversarial examples are about as realistic as benign images.
>
> In addition to the quantitative experiment and the user study, we include some high resolution adversarial examples in Figure 8.

---

> ### Public Comment · (anonymous) · 2019-07-11
> **usefulness of fast generating adversarial examples**
>
> adversarial training needs adversarial examples. Hence fastly generating high-quality adversarial examples may accelerate the adversarial training.

---

### Official Review · AnonReviewer1 · 2017-11-29

**Rating:** 6
**Confidence:** 4

**Review:**

This paper describes AdvGAN, a conditional GAN plus adversarial loss. AdvGAN is able to generate adversarial samples by running a forward pass on generator. The authors evaluate AdvGAN on semi-white box and black box setting.

AdvGAN is a simple and neat solution to for generating adversary samples. The author also reports state-of-art results.

Comment:

1. For MNIST samples, we can easily find the generated sample is a mixture of two digitals. Eg, for digital 7 there is a light gray 3 overlap. I am wondering this method is trying to mixture several samples into one to generate adversary samples. For real color samples, it is harder to figure out the mixture.
2. Based on mixture assumption, I suggest the author add one more comparison to other method, which is relative change from original image, to see whether AdvGAN is the most efficient model to generate the adversary sample (makes minimal change to original image).

---

> ### Author Response · Authors · 2018-01-04
> **Reply to review**
>
> We thank the reviewer for the thoughtful comments and suggestions. We plot the perturbation in Figure 3 (c) (d) and 4 (b) in the updated version. From these plots, we can see that AdvGAN’s perturbations (amplified by 10×) do not resemble images from CIFAR-10/ImageNet.
> For fair comparison against other attacks, we limit the perturbation to 0.3 L_infinity distance for MNIST and 8 for CIFAR-10.
>
> We have compared the attack success rate of adversarial examples by different methods under defenses in Tables 3 and 4 and show that AdvGAN can often achieve high attack success rate under the same perturbation budget compared to other methods.
> We have also added another comparison on MNIST and CIFAR-10 with FGSM and optimization methods, showing the relative change from original image in Table 7 in the appendix.
> From the table, we can see that AdvGAN adds comparable perturbation with CW and less perturbation compared with FGSM. As AdvGAN aims to generate photo realistic images with bounded perturbation instead of minimizing the perturbation as CW does, the perturbation added by AdvGAN is slightly higher compared to CW.

---

### Official Review · AnonReviewer3 · 2017-11-30
**adversarial adversial example generation, wins MadryLab's mnist challenge**

**Rating:** 7
**Confidence:** 3

**Review:**

The paper proposes a way of generating adversarial examples that fool classification systems.
They formulate it for a blackbox and a semi-blackbox setting (semi being, needed for training their own network, but not to generate new samples).

The model is a residual gan formulation, where the generator generates an image mask M, and (Input + M) is the adversarial example.
The paper is generally easy to understand and clear in their results.
I am not awfully familiar with the literature on adversarial examples to know if other GAN variants exist. From this paper's literature survey, they dont exist.
So this paper is innovative in two parts:
- it applies GANs to adversarial example generation
- the method is a simple feed-forward network, so it is very fast to compute

The experiments are pretty robust, and they show that their method is better than the proposed baselines.
I am not sure if these are complete baselines or if the baselines need to cover other methods (again, not fully familiar with all literature here).

---

> ### Author Response · Authors · 2018-01-04
> **Reply to "adversarial adversial example generation, wins MadryLab's mnist challenge"**
>
> We thank the reviewer for the thoughtful comments and suggestions. As mentioned by the reviewer, for baselines comparison, the proposed AdvGAN is currently the best attack method  in Madry et al.’s MNIST Adversarial Examples Challenge (https://github.com/MadryLab/mnist_challenge), which includes many state-of-the-art attack methods.
> In our updated version, we have also added another comparison on MNIST and CIFAR-10 with FGSM and optimization methods, showing the perturbation amount in Table 7 in the appendix.

---

### Public Comment · (anonymous) · 2017-11-08
**GANs for Detecting Adversarial Examples**

If a GAN can learn to attack, can't another GAN learn the adversarial perturbations and defend against it?

---

> ### Author Response · Authors · 2017-11-18
> **Reply to "GANs for Detecting Adversarial Examples"**
>
> It doesn’t follow so easily. We’ve shown that a GAN can generate attacks and that it can generate a variety of adversarial examples, but there’s no evidence that the range of outputs covers the entire space of adversarial examples. Furthermore, GANs only learn to approximate a true distribution based on limited training data--just like a classifier in this respect--so they may be susceptible to adversarial examples in the same way.

---

> > ### Public Comment · (anonymous) · 2017-11-20
> > **GANs for Detecting GAN-Based Adversarial Examples**
> >
> > My question was: can a GAN defend against adversarial examples generated by your method (using GAN)?

---

> > > ### Author Response · Authors · 2017-11-21
> > > **Reply to "GANs for Detecting GAN-Based Adversarial Examples"**
> > >
> > > The answer a very limited yes, where it would appear to work, but may not be a good idea, since this problem is not perfectly symmetric.  It is sufficient to be used as an attack for GAN to generate a few different adversarial examples. However, an efficient defense with GAN has to consider the much broader and more complex space of all adversarial examples.
> > >
> > > For a fixed AdvGAN instance, you should be able to train a discriminator to differentiate the outputs of that specific AdvGAN from benign data. Lee et al. have proposed a related method of using GAN for adversarial training [https://arxiv.org/abs/1705.03387].
> > > However, the resulting discriminator is not very useful as a general defense, because it does not detect other attacks, possibly even another instance of AdvGAN.
> > >
> > > In a similar setting, Carlini & Wagner have shown that the C&W attack can bypass a classifier that’s been trained to detect C&W attacks [https://arxiv.org/abs/1705.07263].

---

> > > > ### Public Comment · (anonymous) · 2017-11-21
> > > > **GAN-based Adversarial Examples Not Effective**
> > > >
> > > > So, I guess this means that adversarial examples generated by GAN can be easier defended compared to other attacks. Essentially, the same GAN that does the attack can do the defense.

---

> > > > > ### Author Response · Authors · 2017-11-22
> > > > > **reply to "GAN-based Adversarial Examples Not Effective"**
> > > > >
> > > > >  I think the meaning of the question has continued to change subtly, with this latest comment bringing up the use of “the same GAN” for doing the defense. What defense method are you thinking of? (Keep in mind that the discriminator in the GAN cannot distinguish the real and generated images by definition of successfully training a GAN.)
> > > > > That aside, the claim that adversarial examples generated by GAN can be easier defended is not the take-away here. Many fixed attacks can easily be mitigated; this would not be unique to AdvGAN. See Carlini & Wagner’s paper [https://arxiv.org/abs/1705.07263] for many such mitigations and simple workarounds that show that they are ultimately ineffective as defenses. We actually show in the paper (table 3,4,5) many cases where adversarial examples generated by AdvGAN are more successful against defenses than other strong attacks. Moreover,  we apply AdvGAN on the MNIST challenge (https://github.com/MadryLab/mnist_challenge) and achieve 88.93% accuracy on the published robust model in the semi-whitebox setting, and 92.76% in the black-box setting, which wins the top position in the challenge. This shows that adversarial examples generated by GAN are actually harder to defend compared with other attacks.

---

> > > > > > ### Public Comment · (anonymous) · 2017-11-27
> > > > > > **Easier to Defend Against Adversarial Example Generator**
> > > > > >
> > > > > > When proposing an attack, we need to think about the right defense. I think we agree that a fixed adversarial example generator can be defended by training a discriminator. Your point that that discriminator cannot defend against other attacks is irrelevant, because here we are talking about defending against your attack, not others'.
> > > > > >
> > > > > > To summarize, while defending against your attacks seems straightforward, this is not the case for other attacks.

---

> > > > > > > ### Author Response · Authors · 2017-11-29
> > > > > > > **reply to "Easier to Defend Against Adversarial Example Generator"**
> > > > > > >
> > > > > > > We do think about defense when proposing an attack. In this paper, we tested our attack on defended models in the evaluation section. In our opinion, the results show that our attack is challenging to defend against because it successfully attacks different kinds of defenses.
> > > > > > >
> > > > > > > To the above commenter, your enthusiasm is appreciated, but we don’t see a ‘straightforward’ way to defend against this attack. It would be helpful if you can provide a proposed defense algorithm for it. We also plan to open source our attack code. Again, our attack was ranked number 1 on the MNIST challenge by Madry’s group, which is a state-of-the-art defense. In our opinion, this suggests that it is not *straightforward* to defend against.

---

> > > > > > > > ### Public Comment · (anonymous) · 2017-11-29
> > > > > > > > **Evaluation Against Appropriate Defense Missing**
> > > > > > > >
> > > > > > > > You did evaluate your attack against different defense methods. But, the question is what would be the *appropriate* defense against your attack? And have you tested your attack against that?

---

> > > > > ### Public Comment · (anonymous) · 2017-11-22
> > > > > **Question about the "easier defended"**
> > > > >
> > > > > Trying to follow this discussion. What was it that makes adversarial examples generated by GAN easier defended compared to other attacks?

---

### Public Comment · (anonymous) · 2017-12-04
**Targeted attacks in the current framework**

In the architectures described in Fig 1 and Appendix B there seems to be no provision for conditioning the adversarial examples generated on the target label. I don't quite understand how you could generate targeted adversarial examples for a test image without providing the target label as an input to the generator. Thanks in advance for your answer.

---

> ### Author Response · Authors · 2018-01-09
> **Reply for "Targeted attacks in the current framework "**
>
> We thank the commenter for the question!
> We are performing targeted attack where we input the target in the loss function as shown in function eq 2, t denotes our target, and we train a generator for a specific target. During test, the trained network will generate the targeted attack for any test image.

---

### Author Response · Authors · 2018-01-08
**Summary of changes to the manuscript**

Changes made in our revised version are listed as below：
- Added perturbation plot in Figure 3(c)(d) and Fig 4(b) for CIFAR-10 and ImageNet, respectively.
- Added a comparison between AdvGAN, FGSM, and optimization methods, comparing the relative changes from original images to adversarial examples (Table 7 in the appendix).
- Added a detailed description in section 3.2 about how to control distortion amount generated by AdvGAN.
- Added suggested references and updated section 2 to include more comprehensive analysis for related work.
- Updated the wording throughout the paper to make it more clear.
- Added human perceptual study and quantitative results for AdvGAN on high resolution images in Section 4.5
- Added additional generated adversarial examples by AdvGAN on high resolution images in Figure 8 in Appendix.

We would like to thank the reviewers again for the useful feedbacks and suggestions.

---

### Decision · Program_Chairs · 2018-01-29
**ICLR 2018 Conference Acceptance Decision**

**Decision:**

Reject

**Comment:**

The paper presents AdvGAN: a GAN that is trained to generate adversarial examples against a convolutional network. The motivation for this method is unclear: the proposed attack does not outperform simpler attack methods such as Carlini-Wagner attack. In white-box settings, a clear downside for the attacker is that it needs to re-train its GAN everytime the defender changes its convolutional network.

More importantly, the work appears preliminary. In particular, the lack of extensive quantitative experiments on ImageNet makes it difficult to compare the proposed approach to alternative attacks methods such as (I-)FGSM, DeepFool, and Carlini-Wagner. The fact that AdvGAN performs well on MNIST is nice, but MNIST should be considered for what it is: a toy dataset. If AdvGANs are, as the authors state in their rebuttal, fast and good at generating high-resolution images, then it should be straightforward to perform comprehensive experiments with AdvGANs on ImageNet (rather than focusing on a small number of images on a single target, as the authors did in their revision)?